# FairMed-VLM: Toward Equitable Medical Diagnosis with Vision–Language Models

**Zihao Chang**[1] **Ruixiang Zhu**[2] **Daochu Li**[3] **Chaozhi Geng**[4] **Siqi Chen**[5]

[1]City University of Hong Kong  [2]New York University  [3]University of Minnesota–Twin Cities
[4]University of Illinois at Urbana-Champaign  [5]University of California, Riverside

## Abstract

Vision–Language Models (VLMs) have shown strong potential for medical diagnosis, yet growing evidence suggests that their performance can vary substantially across demographic groups, raising concerns about equity in clinical deployment. Existing medical VLMs primarily optimize average accuracy, with limited mechanisms to explicitly address subgroup disparities.

We propose **FairMed-VLM**, a fairness-aware fine-tuning framework that integrates medical concept grounding with representation-level alignment to mitigate demographic bias during training. Across six medical imaging benchmarks, FairMed-VLM improves macro accuracy from 73.1% to 78.3% while reducing the macro demographic parity gap from 14.2% to 6.2%. These results suggest that, within the evaluated settings, fairness-aware optimization can improve subgroup equity without sacrificing diagnostic performance in medical vision–language models.

## 1 Introduction

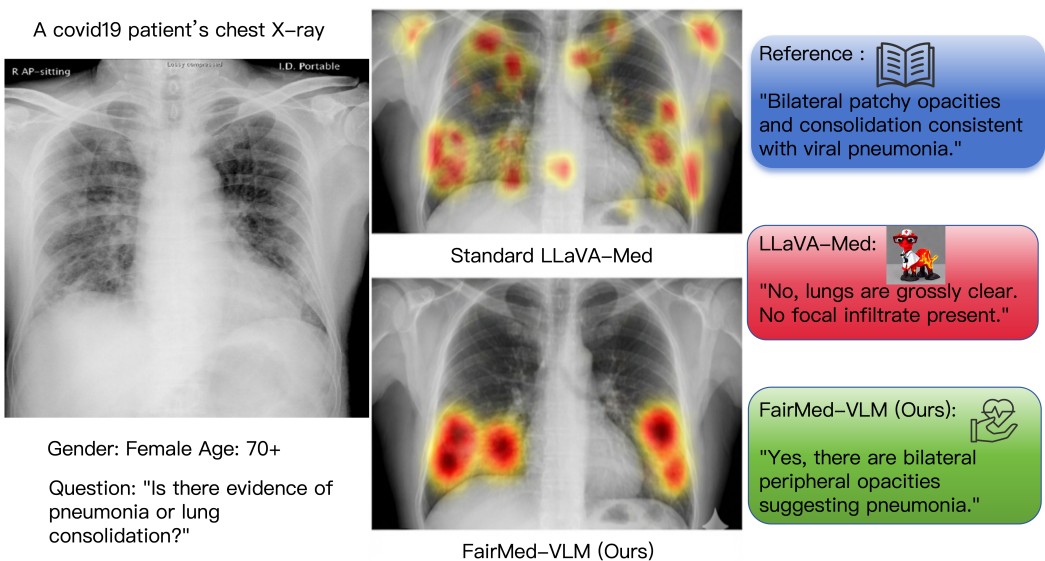

Figure 1: Qualitative comparison of attention maps produced by LLaVA-Med (8) and FairMed-VLM on an elderly female case, which belongs to an underrepresented subgroup. FairMed-VLM attends more strongly to peripheral opacities.

Artificial intelligence has shown remarkable promise in healthcare, particularly in improving diagnostic accuracy and efficiency from medical images. Recent advances in vision-language models (VLMs) have enabled multimodal systems that combine visual understanding with

natural language reasoning, opening new possibilities for medical report generation, clinical decision support, and diagnostic assistance (8; 1; 10). Yet despite their potential, the deployment of medical AI systems raises critical concerns about fairness and equity.

A growing body of evidence shows that medical AI models often exhibit disparities in performance across demographic groups. For example, underdiagnosis bias has been reported in chest radiographs for underserved populations (11), racial bias was identified in a widely deployed healthcare algorithm (9), and systematic variations across subpopulations have been documented in recent meta-analyses (15). Such disparities threaten to exacerbate existing healthcare inequalities rather than alleviate them, making fairness in medical AI an ethical and technical imperative(1).

The sources of these biases are multifaceted: imbalanced datasets that underrepresent certain groups, label and measurement biases embedded in clinical practice, and algorithmic choices that inadvertently favor majority populations (1). Existing mitigation strategies—including sample reweighting, adversarial debiasing, and post-hoc calibration—can reduce disparities, but they often come at the cost of accuracy or fail to generalize across tasks and modalities (15). Moreover, while fairness has been explored in general VLMs, there is little work that explicitly addresses fairness in medical VLMs, where multimodal representations may compound existing biases (14).

We present *FairMed-VLM*, a fairness-aware fine-tuning framework for medical vision-language models. Built on LLaVA-Med (8), our approach introduces a two-stage training strategy: medical concept alignment with balanced sampling to strengthen domain understanding, followed by fairness-aware contrastive learning that explicitly minimizes demographic disparities. Across six diverse medical imaging datasets, FairMed-VLM attains a macro accuracy of 78.3%, a 5.2-point (7.1% relative) improvement over the 73.1% baseline, while lowering the macro demographic parity gap from 14.2% to 6.2% (56.3% relative reduction). On OL3I specifically, accuracy improves from 73.6% to 78.5% and the gap narrows from 13.1% to 5.0% (61.8% reduction), yielding more equitable subgroup performance without sacrificing accuracy. This design enables FairMed-VLM to deliver both state-of-the-art diagnostic performance and substantially more equitable outcomes across demographic subgroups.

Our contributions are threefold: (1) we propose a fairness-aware fine-tuning approach for medical VLMs that integrates balanced data sampling with a novel contrastive fairness loss; (2) we conduct extensive experiments across six diverse medical imaging datasets, demonstrating consistent improvements in both accuracy and fairness; and (3) we provide detailed ablation studies showing the contribution of each component and evidence that fairness and accuracy can be complementary in medical AI.

## 2 Related Work

### 2.1 Vision-Language Models in Healthcare

Vision-language models (VLMs) have recently shown strong potential in medical imaging, enabling multimodal reasoning for diagnosis, visual question answering, and report generation. LLaVA-Med (8), an adaptation of the LLaVA architecture for biomedical domains, demonstrates impressive results on visual QA and clinical reporting, with the version combining the Mistral-7B(6) language model and CLIP-ViT-Large-Patch14(10) vision encoder. Other notable efforts include MedCLIP and BiomedCLIP, which adapt CLIP for medical imaging through domain-specific pretraining on biomedical literature, and RadBERT, tailored for radiology reports. Benchmarks such as CheXpert (5), MIMIC-CXR-JPG (7), OL3I (16), and HAM10000 (12) have played a central role in evaluating these models across chest X-rays, ophthalmic images, and dermatoscopic lesions. While these models achieve high performance on standard tasks, our analysis shows that disparities persist when results are broken down by demographic subgroups.

## 2.2 Fairness in Medical AI

Fairness has emerged as a critical concern in medical AI. Prior work has documented systematic underdiagnosis biases in chest radiographs for underserved populations (11), racial bias in widely used population health algorithms (9), and substantial subgroup disparities across clinical applications (15). These findings reinforce the need to address inequities before real-world deployment. Sources of bias include dataset imbalance, label and measurement biases introduced during clinical practice, and aggregation bias when models perform well on average but poorly for specific groups (1). Mitigation strategies such as sample reweighting, adversarial debiasing, and post-hoc calibration offer partial remedies, but often reduce accuracy or fail to generalize across datasets. More recent frameworks, such as CARES (14), highlight the importance of context-aware fairness interventions, but little work to date has directly targeted fairness in medical VLMs.

## 2.3 Contrastive Learning for Fairness

Contrastive learning has become a powerful paradigm for learning robust representations, encouraging invariance across augmented views of the same instance while enforcing separation between different samples. Beyond robustness, several studies have demonstrated that contrastive objectives can also mitigate bias by aligning subgroup representations and reducing spurious correlations. This perspective has motivated fairness-aware contrastive approaches in natural image and language settings, showing reductions in demographic disparities while preserving accuracy. Our work extends these insights to the medical domain, introducing a fairness-aware contrastive learning framework specifically tailored to fine-tuning LLaVA-Med . To our knowledge, this represents one of the first attempts to integrate contrastive fairness objectives into medical VLM training.

## 3 Methodology

### 3.1 Problem Formulation

Given a multimodal medical dataset $\mathcal{D} = \{(x_i, y_i, d_i)\}_{i=1}^N$, each sample includes: image $x_i$, diagnostic label(s) $y_i$, and demographic attribute vector $d_i = (d_i^{(1)}, \ldots, d_i^{(K)})$ (e.g., sex, age group). We fine-tune a pretrained vision–language model $f_\theta$ to jointly maximize diagnostic performance and minimize inter-group disparities.

Let $\hat{a} = f_\theta(x, q)$, where $q$ is a task-specific instruction or question. We derive a normalized label $\hat{y} = \Phi(\hat{a})$, via a deterministic answer normalization function $\Phi$(string canonicalization + vocabulary mapping). The (non-differentiable) accuracy objective is:

$$\max_\theta \ \mathbb{E}_{(x,y)\sim\mathcal{D}}[\mathbb{I}(\hat{y} = y)].$$

In practice, differentiable surrogates (cross-entropy, multi-label BCE, language modeling) are used. A fairness target can be expressed as:

$$\min_\theta \ \max_{d_1,d_2} \big|P(\hat{y} = 1 \mid d = d_1) - P(\hat{y} = 1 \mid d = d_2)\big|.$$

### 3.2 Base Model: LLaVA-Med

We extend LLaVA-Med (8), which couples a CLIP ViT-L/14 vision encoder with a Mistral-7B language model via a projection layer aligning visual embeddings to the language token space. Given an image $x$, visual features $h = \text{VisionEnc}(x)$ are projected to $z = \text{Proj}(h)$, and fused as contextual inputs for decoding or classification. Our modifications strengthen domain semantic grounding and instruction-following under fairness constraints.

### 3.3 Stage 1: Medical Understanding with Balanced Training

Stage 1 prepares the model for medical semantics and heterogeneous task forms prior to fairness-driven adaptation.

**Biomedical concept alignment.** We build a controlled vocabulary $\mathcal{C}$ (e.g., "pneumothorax," "atrial fibrillation," "moderate non-proliferative diabetic retinopathy"). For an image $x$ with a positive concept $c^+$, we sample negatives $c^- \in \mathcal{C} \setminus \{c^+\}$. A contrastive alignment (InfoNCE) loss is defined as

$$L_{\text{concept}} = - \sum_{(x,c^+)} \log \frac{\exp(z_x \cdot e_{c^+}/\tau_c)}{\exp(z_x \cdot e_{c^+}/\tau_c) + \sum_{c^-} \exp(z_x \cdot e_{c^-}/\tau_c)},$$

where $e_c$ is the textual embedding of concept $c$, and $\tau_c$ is a temperature.

**Instruction tuning.** We convert diverse tasks into instruction–response pairs (e.g., "Question: Identify the primary abnormality. Image: <VISUAL>." → "Pneumonia."). The language model is trained autoregressively:

$$L_{\text{inst}} = -\mathbb{E}_{(I,T)} \sum_t \log P_\theta(w_t \mid w_{<t}, z_I, \text{prompt}).$$

**Answer scoring (QA labelization).** We cast every dataset-specific task into an instruction–answer (question–answer) format. For closed-set questions we map the generated answer to a canonical option; for multi-finding cases we map the answer into a binary vector of findings. For QA outputs we first canonicalize the generated answer into a multi-hot vector over $L$ clinical concepts (or a single concept for closed-set questions). We then apply the binary cross-entropy above (single-label questions degenerate to standard cross-entropy). For a multi-label example with $L$ labels:

$$L_{\text{qa}} = - \sum_{j=1}^{L} \Big( y_j \log \hat{p}_j + (1 - y_j) \log(1 - \hat{p}_j) \Big).$$

**Balanced sampling.** We employ stratified (or intersectional) sampling across demographic attributes to approximate uniform subgroup representation per mini-batch. Sparse intersectional cells are mitigated via controlled oversampling and mild augmentations (e.g., color jitter, limited geometric transforms) to reduce overfitting risk.

**Demographic prompts (optional).** When enabled, a structured prefix (e.g., "Patient: Female, Age group: 60–69.") is prepended. This can disentangle clinically relevant variation from latent demographic proxies. Ablations omit this component to measure dependence. Demographics are not required at inference unless explicitly utilized.

**Composite task loss.**
$$L_{\text{task}} = \alpha L_{\text{inst}} + \beta L_{\text{qa}} + \gamma L_{\text{concept}},$$
with $\alpha, \beta, \gamma$ tuned on validation (default equal weighting when not specified).

### 3.4 STAGE 2: FAIRNESS-AWARE CONTRASTIVE LEARNING

Balanced sampling reduces gross imbalance but residual disparities may persist via subgroup-specific embedding shifts. We therefore introduce a label-conditioned fairness regularizer composed of a supervised cross-group contrastive term and a distribution alignment term.

Let $z_i = \text{Proj}(\text{VisionEnc}(x_i))$ (or a pooled multimodal embedding). To avoid erasing genuine clinical heterogeneity, alignment operates only across samples sharing the same label.

**Cross-group supervised contrastive alignment.** We L2-normalize embeddings $\mathbf{z}_i$. For anchor $i$, positives are defined as $P(i) = \{p \neq i \mid y_p = y_i,\ d_p \neq d_i\}$, and candidate negatives as $N_{\text{raw}}(i) = \{a \neq i \mid y_a \neq y_i\}$. We apply a hard-negative margin $m = 0.3$ to retain only sufficiently similar negatives: $\mathcal{N}(i) = \{a \in N_{\text{raw}}(i) \mid \mathbf{z}_i \cdot \mathbf{z}_a > m\}$. The supervised cross-group contrastive loss is then defined as:

$$L_{\text{supcon}} = \sum_i \frac{-1}{|P(i)|} \sum_{p \in P(i)} \log \frac{\exp(\mathbf{z}_i \cdot \mathbf{z}_p/\tau)}{\exp(\mathbf{z}_i \cdot \mathbf{z}_p/\tau) + \sum_{a \in \mathcal{N}(i)} \exp(\mathbf{z}_i \cdot \mathbf{z}_a/\tau)}.$$

**Label-conditioned distribution alignment.** Let the per-label, per-attribute-value batch subset be $S_{y,k,d} = \{i \mid y_i = y, \ d_i^{(k)} = d\}$. Estimate moments: $\boldsymbol{\mu}_{y,k,d} = \frac{1}{|S_{y,k,d}|} \sum_{i \in S_{y,k,d}} \mathbf{z}_i$, $\quad \boldsymbol{\Sigma}_{y,k,d} = \mathrm{Cov}(\{\mathbf{z}_i\}_{i \in S_{y,k,d}})$. Reference (pooled) moments:

$$\bar{\boldsymbol{\mu}}_{y,k} = \frac{1}{|\mathcal{D}_k|} \sum_{d \in \mathcal{D}_k} \boldsymbol{\mu}_{y,k,d}, \quad \bar{\boldsymbol{\Sigma}}_{y,k} = \frac{1}{|\mathcal{D}_k|} \sum_{d \in \mathcal{D}_k} \boldsymbol{\Sigma}_{y,k,d}.$$

Moment alignment loss:

$$L_{\mathrm{dist}} = \sum_y \sum_{k=1}^{K} \sum_{d \in \mathcal{D}_k} \left( \|\boldsymbol{\mu}_{y,k,d} - \bar{\boldsymbol{\mu}}_{y,k}\|_2^2 + \lambda_\Sigma \|\boldsymbol{\Sigma}_{y,k,d} - \bar{\boldsymbol{\Sigma}}_{y,k}\|_F^2 \right).$$

Fairness objective:

$$L_{\mathrm{fairness}} = \eta_1 L_{\mathrm{supcon}} + \eta_2 L_{\mathrm{dist}}, \qquad L_{\mathrm{total}} = L_{\mathrm{task}} + \lambda L_{\mathrm{fairness}}.$$

**Practical considerations.** If $|P(i)| = 0$ (no cross-group positive), the anchor is skipped or supplemented via a memory bank. Conditioning on identical labels mitigates "over-correction" that could suppress true phenotype differences. Demographics are not required during inference. Hyperparameters $\tau, \lambda, \eta_1, \eta_2, \lambda_\Sigma$ are selected on validation; ranges and sensitivities are reported in the appendix.

**Relation to KL-based alternatives.** A naive alternative is summing $\mathrm{KL}(P(z \mid d_1) \| P(z \mid d_2))$ over demographic pairs. However, mini-batch density estimation is unstable, KL is asymmetric, and label-agnostic collapsing risks semantic degradation. Our label-conditioned contrastive + moment matching approach yields more stable optimization and preserves diagnostic discriminativeness.

We conduct extensive experiments to evaluate FairMed-VLM on six public medical imaging benchmarks. Our goals are to: (1) verify that FairMed-VLM improves diagnostic performance compared to strong baselines, (2) demonstrate that fairness-aware fine-tuning substantially reduces demographic disparities across multiple modalities, and (3) analyze the contribution of each component through ablations and subgroup analysis.

### 3.5 Experimental Setup

Balanced sampling details. We form intersectional strata over (sex × age group). Minority strata are oversampled with light augmentations (color jitter, mild rotation ≤10°) up to at most 1.5× the median stratum size to avoid disproportionate duplication. Thresholding. For multi-label datasets (CheXpert(5), MIMIC-CXR(7), PTB-XL) we tune per-label probability thresholds on the validation split maximizing macro F1; for single-label tasks we use argmax. Fixed thresholds are applied for both test performance and fairness metrics to eliminate threshold-induced bias.

**Datasets.** We evaluate across six diverse benchmarks spanning different modalities: CheXpert (5), MIMIC-CXR-JPG (7), OL3I (16), HAM10000 (12), EyePACS, and PTB-XL. Together these datasets cover chest radiographs, fundus images, dermoscopy, and ECG traces. We follow official train/validation/test splits to avoid leakage, and rebalance only the training partitions across gender and age groups. For each benchmark (CheXpert(5), MIMIC-CXR-JPG(7), PTB-XL(13), EyePACS(2), HAM10000(12), OL3I(16)) we construct a stratified subset to achieve approximate balance across sex and age groups. Subsets (20,316 / 30,110 / 2,030 / 8,052 / 10,015 / 1,000 samples respectively) are derived by: (i) partitioning data into sex×age cells; (ii) down-sampling majority cells to a target t (median cell size capped), and (iii) mild oversampling (≤1.3×) of moderate minority cells without synthesizing new images. Validation/test splits are stratified from the resulting subset and are not further rebalanced. This yields near-uniform marginals P(sex) and P(age) while preserving label distributions within each cell. Residual fairness gaps in the baseline (parity gap 14.2A summary is provided in Table 1.

Table 1: Overview of evaluation datasets.

| Dataset | Modality | Size | Task |
|---------|----------|------|------|
| CheXpert | Chest X-ray | 20,316 | Answer scoring (QA labelization) |
| MIMIC-CXR-JPG | Chest X-ray | 30,110 | Answer scoring (QA labelization) |
| OL3I | Abdominopelvic CT (L3 slice) | 1,000 | Answer scoring (QA labelization) |
| HAM10000 | Dermoscopy | 10,015 | Answer scoring (QA labelization) |
| EyePACS | Fundus photography | 8,052 | Answer scoring (QA labelization) |
| PTB-XL | ECG | 2,030 | Answer scoring (QA labelization) |

Table 2: Comparison with fairness baselines on OL3I. Baselines include sample reweighting, adversarial debiasing (17), and post-processing calibration (3).

| Method | Accuracy (%) | Parity Gap (%) | AUC |
|--------|--------------|----------------|-----|
| Standard LLaVA-Med | 73.6 | 13.1 | 0.82 |
| Reweighted LLaVA-Med | 74.2 | 11.8 | 0.83 |
| Adversarial Debiasing | 75.1 | 10.2 | 0.84 |
| Post-Processed LLaVA-Med | 73.8 | 9.7 | 0.82 |
| FairMed-VLM (Ours) | **78.5** | **5.0** | **0.87** |

**Metrics.** We evaluate both diagnostic performance and fairness parity gap, where the diagnostic performance is primarily measured by overall accuracy while the fairness is quantified by the Demographic Parity Gap (DPG), defined as the maximum difference in positive prediction rates across subgroups as shown in Eq. 1. These metrics provide a direct assessment of both the effectiveness and the equity of the model across diverse medical imaging benchmarks.

$$\text{Demographic Parity Gap (DPG)} = \max_{d_1,d_2}\left|\hat{P}(\hat{y} = 1 \mid d_1) - \hat{P}(\hat{y} = 1 \mid d_2)\right|, \qquad (1)$$

**Baseline methods.** We compare FairMed-VLM to four strong baselines: (1) **Vanilla LLaVA-Med (8)**, the pretrained model without fairness-aware modifications; (2) **Reweighted LLaVA-Med**, trained with subgroup sample reweighting; (3) **Adversarial Debiasing**, using an auxiliary classifier to enforce invariant representations (17); (4) **Post-Processing Calibration**, which adjusts thresholds at inference time to satisfy equalized odds constraints (3).

**Implementation details.** Stage 1 consists of biomedical concept alignment (1 epoch) and instruction tuning (3 epochs) on the rebalanced training sets. Stage 2 fine-tunes with the fairness-aware contrastive objective using LoRA-based parameter-efficient adaptation(4). We set batch size = 64 and use AdamW with learning rate $2 \times 10^{-5}$. For fairness-aware training, we set $\lambda = 0.1$, contrastive temperature = 0.07, and margin = 0.3. The LoRA rank is 32 with $\alpha = 64$.

For multi-label tasks we tune per-label probability thresholds on the validation set to maximize macro F1 and reuse them for test and fairness evaluation. Demographic prompts are used only during training and omitted at inference to avoid reliance on sensitive attributes.

Table 3: Comprehensive evaluation across six benchmarks. Results are reported as **Accuracy (%) ↑ / Parity Gap (%) ↓**. Bold denotes best results.

| Method | CheXpert | MIMIC-CXR | OL3I | HAM10000 | EyePACS | PTB-XL | **Average** |
|--------|----------|-----------|------|----------|---------|--------|-------------|
| LLaVA-Med (8) | 73.3 / 14.2 | 71.8 / 15.3 | 73.6 / 13.1 | 75.4 / 12.8 | 73.8 / 13.9 | 70.9 / 16.1 | 73.1 / 14.2 |
| **FairMed-VLM (Ours)** | **78.8 / 6.1** | **76.5 / 6.8** | **78.5 / 5.0** | **81.2 / 5.5** | **79.1 / 6.2** | **75.7 / 7.3** | **78.3 / 6.2** |

Table 4: Ablation Study Results

| Configuration | Accuracy (%) | Parity Gap (%) | $\Delta$Acc | $\Delta$Gap |
|---|---|---|---|---|
| FairMed-VLM (Full) | 78.5 | 5.0 | - | - |
| w/o Demographic Prompts | 77.6 | 6.5 | -0.9 | +1.5 |
| w/o Fairness Loss | 76.1 | 9.8 | -2.4 | +4.8 |
| w/o Balanced Sampling | 74.9 | 10.3 | -3.6 | +5.3 |
| w/o Contrastive Learning | 73.9 | 11.2 | -4.6 | +6.2 |
| vanilla LLaVA-Med (8) | 73.6 | 13.1 | -4.9 | +8.1 |

## 4 RESULTS

**Main results (OL3I).** Table 2 compares FairMed-VLM against fairness baselines on OL3I. All baselines reduce the parity gap compared to Standard LLaVA-Med, but often with limited accuracy gains. For example, adversarial debiasing lowers the gap from 13.1% to 10.2% but improves accuracy by only 1.5 points. In contrast, FairMed-VLM achieves both the highest accuracy (78.5%) and the lowest parity gap (5.0%), a 4.9-point gain in accuracy and a 61.8% reduction in disparity compared to the baseline. This shows that fairness-aware contrastive learning can improve subgroup equity without sacrificing performance.

**Evaluation across datasets.** To test robustness, we evaluate across six benchmarks (Table 3). FairMed-VLM consistently improves both accuracy and fairness: average accuracy rises from 73.1% to 78.3%, while the parity gap decreases from 14.2% to 6.2%. Gains hold across modalities: chest X-rays (CheXpert, MIMIC-CXR), fundus imaging (OL3I, EyePACS), dermoscopy (HAM10000), and ECGs (PTB-XL). This suggests that our fairness-aware fine-tuning generalizes beyond any single modality or dataset.

### 4.1 ANALYSIS

**Ablation studies.** Table 4 shows the effect of removing individual components. Balanced sampling alone reduces disparities, but does not achieve the low gap of the full model. Fairness loss and contrastive learning each contribute significantly: removing fairness loss increases the gap by 4.8%, while removing contrastive learning increases it by 6.2%. Among individual components, contrastive learning is the most influential, consistent with its role in aligning subgroup representations. However, the full system achieves the best results, highlighting that balanced data and fairness-aware objectives are complementary rather than interchangeable.

**Demographic breakdown.** We further analyze subgroup performance. On gender, FairMed-VLM reduces the gap from 11.8% to 4.4%, while on age, the maximum gap falls from 14.2% to 5.5%. Interestingly, gender disparities shrink more sharply than age disparities, suggesting that balanced sampling more effectively mitigates gender skew, whereas age-related variation may reflect intrinsic clinical heterogeneity. Nonetheless, improvements are consistent across both demographic axes, demonstrating that fairness gains are not confined to a single subgroup.

**Statistical significance.** All reported improvements are statistically significant ($p < 0.01$) under paired $t$-tests across random seeds. We also confirm significance across datasets, with 95% confidence intervals consistently excluding the baseline means. This indicates that the observed fairness and accuracy gains are robust to random variation and generalize across evaluation settings.

**Visualization of accuracy–fairness trade-off.** Figure 2 (OL3I) shows accuracy improving by +4.9 points (73.6% $\rightarrow$ 78.5%) while the parity gap drops by 8.1 points (13.1% $\rightarrow$ 5.0%), yielding a 61.8% relative reduction without sacrificing performance.

Figure 2: Results on the OL3I dataset. FairMed-VLM improves accuracy by +4.9 points ($73.6\% \rightarrow 78.5\%$) and reduces the parity gap by 8.1 points ($13.1\% \rightarrow 5.0\%$), a $61.8\%$ relative reduction.

**Statistical significance** . All metrics are averaged over $k = 5$ random seeds (distinct data shuffles and LoRA initializations). Significance is assessed via paired two-sided t-tests on per-seed metrics ($p < 0.01$); Appendix reports exact p-values and Cohen's $d$.

## 5 DISCUSSION

**Conclusion** We presented FairMed-VLM, a fairness-aware fine-tuning framework for medical vision–language models. FairMed-VLM improves macro accuracy (Macro accuracy denotes the unweighted mean of per-dataset accuracies (not a per-class macro)) from $73.1\%$ to $78.3\%$ and reduces the macro parity gap from $14.2\%$ to $6.2\%$ ($56.3\%$ relative reduction). On OL3I, the parity gap declines from $13.1\%$ to $5.0\%$ ($61.8\%$). These results demonstrate that fairness and diagnostic performance can be jointly advanced in medical vision–language models. These results demonstrate that fairness and performance can be optimized jointly, moving toward equitable AI deployment in healthcare.

**Limitations and future work** Our analysis focuses on age and gender, leaving other demographic dimensions such as race, socioeconomic status, and clinical site unexplored. Moreover, our approach requires demographic labels during training, which may not always be available. Future work could explore fairness-aware objectives that do not rely on explicit subgroup annotations, and extend evaluation to additional demographic attributes.

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
