# OpenReview forum: "FairMed-VLM: Toward Equitable Medical Di- agnosis with Vision–Language Models"
_ICLR.cc/2026/Workshop/AFAA — AFAA 2026 Poster_

### Official Review · Reviewer_cTcE · 2026-02-14
**This paper presents original work that closely aligns with the workshop's purpose. However, it still needs substantial work, especially to improve its readability and better support its main claim of offering a new approach that avoids an accuracy-fairness trade-off.**

**Rating:** 3
**Confidence:** 4

**Summary:**

This paper proposes adapting the LLAVA-MED model by fine-tuning it with two objectives. The first aims to improve the model's understanding of medical semantics and its generalization to heterogeneous medical tasks by adding a medical vocabulary, a new prompting approach, and balanced sampling. The second objective function aims at improving fairness through contrastive learning. The paper demonstrates that this new approach improves both accuracy and group parity for VLM applied to medical tasks.

**Strengths:**

For an algorithmic fairness workshop, this paper is well-positioned. Specifically, the paper addresses an important aspect that is often overlooked: it explicitly highlights subgroup disparities in clinical AI deployment. It shifts the fairness discussion from generic image models to a clinically consequential multimodal setting.

The paper proposes a good framing of the problem and clearly articulates the fairness gap in medical VLMs. Moreover, it is well-structured and proposes robust evaluations across multiple datasets, with several baselines and an ablation study to understand the role of the model's components.

The paper has clear conceptual contributions. It proposes a two-stage training approach to improve an existing model: a medical-domain strengthening approach to improve the model's accuracy and a fairness-aware contrastive alignment to reduce group disparity.

The paper presents promising results as the overall method comes with fairness improvements without an accuracy trade-off.

The paper has good discussion potential. It could stimulate discussion of representation-level fairness and the feasibility of reducing group disparities in VLMs without sacrificing accuracy.

**Weaknesses:**

**General comments:**
- The claim on the Fairness-Accuracy trade-off: As it stands, I do not agree with the general claim that they identified a fairness mitigation approach with no trade-off for accuracy. While the model's overall performance does improve both accuracy and fairness, the paper combines two approaches: one to improve medical understanding and a contrastive learning approach to increase fairness. The methodology section is currently not self-contained and is extremely difficult to interpret. Core notations are introduced without definition, several technical terms are assumed rather than explained, and the objective function is presented without sufficient conceptual motivation. Even after multiple readings, I was unable to clearly reconstruct what the model is optimizing or how the components interact. Once again, I will provide detailed examples just below. I provide more details below on what should be added to the ablation study to support my point.
- The related work section on fairness in medical AI is almost a copy-and-paste of what is already written in the introduction. I think that more details on the various approaches you used for your baseline should be provided here, for example, and then compared to your approach.

**Methodology:**
- Motivation behind each component: What is the philosophy behind Biomedical concept alignment, Instruction tuning, or Cross-group supervised contrastive alignment. Why are you approaching the problem this way, and what is this supposed to do in your model? These are essential information to improving the understanding of your paper.
- Technical terms: What is a negative sample with negative concept in the biomedical alignment section? What is a canonical option or a binary finding in the Question Answering section? What are positives and negatives in the contrastive learning section? What is the memory bank and what are estimate moments? All these terms, without definitions or concrete examples of what they correspond to, make understanding the section very difficult.
- Notations: Methodology lacks notations. I will just use the instruction tuning as an example, but you can generalize my point to the whole section. So, in the equation for L_inst, we introduce the terms I and T, which I assume are sets of terms and images? I am not even sure, because in the introduction, the author states, "Given an image x." Then what is Zi in the notation, and why is it not Zx anymore, as in the previous section? What are the wt? Are these the words of the input prompts? All of these notations need proper definitions to be able to understand the equation properly, especially when we don't know what components are supposed to do.

**Experimental setup:**
- Reproducibility: The authors did not provide any details on the release of their code publicly, which I assume they will do to ensure reproducibility of their results. However, they used a wide range of sample sizes, thresholds, and other hyperparameters without providing their values or the rationale behind them when they were provided. For example, in the Balanced sampling details section, how do you justify the use of these parameters for data augmentation? Is this a standard practice? Can you provide a reference? Additionally, what values are the thresholds you are using after the optimization for each dataset? Even if these parameters could be included in the article's appendix, they should still be mentioned.
- Analysis/Ablation Study: The ablation study misses removing some components of the approach to be complete. In particular, the authors did not perform ablation for the Biomedical concept alignment, Instruction tuning, or Answer scoring (QA labelization) parts of the models. More importantly, the ablation should remove the entire stage 1 and stage 2 from the performance to assess whether the model really addresses the accuracy/fairness trade-off. In my understanding, for now, stage 1 addresses accuracy, and stage 2 addresses fairness; the combination of the two means there is no trade-off. If this is confirmed it would mean that you did not identify a fairness approach that balance trade-off.

**Other remarks:**
- Relation to KL-based alternatives: KL-based alternatives are not used as baselines in the remainder of the paper. Therefore, it should not be presented in the methodology section but rather in the related work section, with the necessary references. If the authors consider it essential to the methodology, they should provide appropriate results for this part.
- Metrics: Can the authors provide references for the Demographic Parity Gap (DPG). I also suggest the authors to provide rationale for the choice of this specific metrics instead of studying equality of odds or opportunity for example.
- Datasets: The references for EyePACS, and PTB-XL should be provided on line 252.
- Main results: I have trouble understanding why the main results focus only on the OL3I datasets, given that the authors compared all datasets. We wonder why the authors did not apply all baseline models to all datasets to enable a fair comparison for all these cases.
- Ablation Study: It is unclear what the authors mean by "contrastive learning" in this section and the discussion. Is it the Cross-group supervised contrastive alignment part of the model, or is it the full stage 2 process? The same question could be asked about what is called the "fairness loss". Is it the label-conditioned distribution alignment of the full stage 2 this time?

---

### Official Review · Reviewer_1dox · 2026-02-18
**This paper proposes a two-stage fairness-aware fine-tuning strategy for medical vision–language models that aligns representations across demographic groups while preserving diagnostic accuracy. Experiments on six datasets show consistent improvements in both accuracy and demographic parity. The approach is technically solid, though some aspects of statistical and subgroup reporting could be clarified.**

**Rating:** 5
**Confidence:** 4

**Summary:**

This paper studies performance differences across demographic subgroups in medical vision–language models and proposes a fine-tuning strategy to reduce these gaps. Building on LLaVA-Med, the authors design a two-stage training procedure. In the first stage, the goal is to strengthen the model’s medical understanding. They do this through medical concept alignment, supervised answer training, instruction-based formatting, and balanced sampling across demographic groups such as age and gender.

In the second stage, the goal is to reduce unfair differences between groups. The model is trained to produce similar internal representations for patients who share the same diagnosis but belong to different demographic groups. This is done using a supervised contrastive objective and by aligning the statistical properties (such as means and variances) of representations across subgroups. Importantly, this alignment is applied only within the same diagnosis so that real clinical differences are not removed.

Experiments conducted on six medical imaging datasets, with comparisons against four baseline methods, show that the proposed approach consistently reduces demographic parity gaps while improving overall predictive performance. These results suggest that fairness constraints can be integrate into medical VLM training without sacrificing accuracy.

**Strengths:**

1. The paper applies fairness-aware contrastive learning to medical vision–language models. While contrastive learning and fairness methods have been studied before, combining them at the representation level in medical VLMs is relatively new and relevant.

2. The two-stage training procedure is clearly structured. The first stage strengthens medical understanding through concept alignment, instruction supervision, and balanced sampling. The second stage introduces a fairness objective that aligns representations across demographic groups while conditioning on the diagnosis label, which helps avoid removing clinically meaningful differences.

3. The method is evaluated on six medical datasets covering different modalities. The authors compare against four baseline approaches and report consistent improvements in both accuracy and demographic parity.

4. The ablation experiments systematically evaluate each component of the framework and show how performance and fairness change when individual elements are removed. The authors also report statistical significance tests, which add support to the robustness of the observed improvements.

5. The paper is concise and well organized. The methodology is clearly described, and the connection between design choices and experimental outcomes is generally easy to follow.

**Weaknesses:**

Although the paper makes solid contributions, clarifying a few points could further strengthen it.
1. The paper reports paired t-tests over five random seeds. With such a small number of runs, the assumption behind the t-test is difficult to verify. Using a non-parametric alternative such as a sign test, or reporting the mean and variance for each metric, would provide more robust evidence.
2. The paper mainly reports overall accuracy together with demographic parity gap (e.g., Tables 2 and 4). It would be helpful to also report subgroup-specific accuracy to better understand how performance varies across groups. In addition, including other baseline methods in 3. In some tables, it is not clear whether the reported metric corresponds to a specific dataset or an average across datasets. Clearer labeling would improve readability.
4. The grouping strategy for age is not fully described. Since age is a continuous variable, specifying how age bins were defined would improve transparency and reproducibility.

---

### Official Review · Reviewer_6d1w · 2026-02-24
**Introducing Demographic Equity to Medical VLMs via Balanced Sampling and Contrastive Learning**

**Rating:** 4
**Confidence:** 4

**Summary:**

This paper tackles the issue of fairness in medical Vision Language Models (VLMs). In particular, the authors propose to enforce outcome equity among demographic subgroups (age, sex). To this end, they adopt the LLaVA-Med model (which is a VLM for medical data) and extend it with a two-stage training strategy: domain understanding (medical concept alignment and balanced sampling) and minimizing demographic disparity (via contrastive learning). The resulting model is evaluated on macro accuracy (averaged over datasets) and demographic parity gap (DGP, maximal difference between a pair of demographics). This approach is shown to beat four other variations of the LLaVA-Med model on one dataset, as well as vanilla LLaVA-Med on six datasets (both in terms of macro accuracy and DGP). The paper concludes with and analysis of the results and a discussion of future work.

**Strengths:**

The paper addresses an important question of fairness in medical VLMs.


The proposed method achieves its goal of demographic equity. It is novel and technically sound. The experiments show that the proposed approach consistently beats the baseline LLaVA-Med model on both accuracy and DGP.


The paper is well-written and easy to follow.

**Weaknesses:**

The main weakness of the paper is the lack of comparison (theoretical or experimental) with other fairness-oriented medical VLM approaches (contrastive-learning-based or not). The authors even go as far as to claim that no such methods exist (lines 67, 113, 125) but this is not true, see [1,2,3] or even models that use a similar definition of distributional equity across demographics but with the Sinkhorn loss [4]. I believe that the particular algorithm proposed by the authors (balanced sampling + contrastive learning) is novel, but the paper would benefit from a discussion of the other methods or ideally an experimental comparison.



As far as experimental evaluation is concerned, the only comparison to other models present in the paper (four variations of the LLaVA-Med baseline) is only performed on one dataset (OL3I, Table 2), as opposed to using all 6 data sets for comparison with vanilla LLaVA-Med (Table 3). Especially in the absence of a comparison with other fairness-oriented medical VLM approaches (see above), seeing how the four variations from Table 2 fare on all six datasets would make the evaluation more convincing.




Another limitation of the paper is the rather simplistic definition of fairness: it is implicitly assumed to be demographic distributional equity of outcome, DPG (line 143, Equation (1)). While this assumption facilitates modeling (and is used in other papers), other definitions might be worth examining too, e.g. equality of odds might be able to accommodate for possible heterogeneity within labels and telling apart non-medical bias from variation within a particular label. E.g. [3] examines more metrics of fairness.



[1] Wang, Peiran et al. Fair-MoE: Fairness-Oriented Mixture of Experts in Vision-Language Models 2025


[2] Ruinan Jin et al. FairMedFM: Fairness Benchmarking for Medical Imaging Foundation Models, 2024


[3] Luo et al. FairCLIP: Harnessing Fairness in Vision-Language Learning, 2024


[4] Bansal et al. Robust Fairness Vision-Language Learning for Medical Image Analysis, 2025




Typo:


l. 175 above $\rightarrow$ below

---

### Meta-Review · Area_Chair_aZbP · 2026-02-26

**Recommendation:** Main Papers Track
**Confidence:** 5

**Metareview:**

The paper targets fairness in medical VLMs by extending LLaVA-Med with a two-stage training pipeline. Reviewers find the problem important and the results promising across six datasets, showing improved accuracy and reduced demographic parity gaps. Key concerns are about positioning, baselines, and clarity: the paper should better situate itself relative to prior fairness-oriented medical VLM/VL work, expand comparisons, and more clearly justify the fairness objective/metrics. One reviewer also flags substantial readability/reproducibility issues (undefined terms/notation, missing hyperparameter details) and requests ablations that more cleanly separate Stage 1 vs. Stage 2 to support claims about avoiding an accuracy–fairness trade-off. Overall, with two accepts and one borderline, I lean toward acceptance.

---

### Decision · Program_Chairs · 2026-03-02

Accept (Poster)